# The Use of Autologous Chondrocyte and Mesenchymal Stem Cell Implants for the Treatment of Focal Chondral Defects in Human Knee Joints—A Systematic Review and Meta-Analysis

**DOI:** 10.3390/ijms23074065

**Published:** 2022-04-06

**Authors:** Ilias Ektor Epanomeritakis, Ernest Lee, Victor Lu, Wasim Khan

**Affiliations:** 1School of Clinical Medicine, University of Cambridge, Cambridge CB2 0SP, UK; iee21@cam.ac.uk (I.E.E.); echl2@cam.ac.uk (E.L.); victorluwawa@yahoo.com.hk (V.L.); 2Department of Trauma and Orthopaedic Surgery, Addenbrooke’s Hospital, University of Cambridge, Cambridge CB2 0QQ, UK

**Keywords:** autologous chondrocyte implantation, mesenchymal stem cells, cartilage, femoral condyle, implant, scaffold, cartilage to cartilage integration

## Abstract

Focal chondral defects of the knee occur commonly in the young, active population due to trauma. Damage can insidiously spread and lead to osteoarthritis with significant functional and socioeconomic consequences. Implants consisting of autologous chondrocytes or mesenchymal stem cells (MSC) seeded onto scaffolds have been suggested as promising therapies to restore these defects. However, the degree of integration between the implant and native cartilage still requires optimization. A PRISMA systematic review and meta-analysis was conducted using five databases (PubMed, MEDLINE, EMBASE, Web of Science, CINAHL) to identify studies that used autologous chondrocyte implants (ACI) or MSC implant therapies to repair chondral defects of the tibiofemoral joint. Data on the integration of the implant-cartilage interface, as well as outcomes of clinical scoring systems, were extracted. Most eligible studies investigated the use of ACI only. Our meta-analysis showed that, across a total of 200 patients, 64% (95% CI (51%, 75%)) achieved complete integration with native cartilage. In addition, a pooled improvement in the mean MOCART integration score was observed during post-operative follow-up (standardized mean difference: 1.16; 95% CI (0.07, 2.24), *p* = 0.04). All studies showed an improvement in the clinical scores. The use of a collagen-based scaffold was associated with better integration and clinical outcomes. This review demonstrated that cell-seeded scaffolds can achieve good quality integration in most patients, which improves over time and is associated with clinical improvements. A greater number of studies comparing these techniques to traditional cartilage repair methods, with more inclusion of MSC-seeded scaffolds, should allow for a standardized approach to cartilage regeneration to develop.

## 1. Introduction

Globally, osteoarthritis of the knee is a leading cause of disability, exacerbated by an aging population and the rising prevalence of obesity [1]. The disease incurs substantial economic costs both to healthcare systems and to individual patients [2]. Its progressive nature frequently necessitates joint replacements. A therapy that avoids knee replacement would therefore save significant costs [3,4]. Conservative measures aimed at preventing the need for surgery show mixed results. Exercise improves pain and function, but this is short-lived [5]. Intra-articular analgesics, corticosteroids, hyaluronic acid, autologous serum and platelet rich plasma cannot mitigate the progressive inflammatory process involved [6].

Traumatic injury to the articular cartilage risks the development of knee osteoarthritis at a younger age [7]. Although established osteoarthritis is diffuse, the disease may develop from cartilage surrounding a focal lesion. The presence of a focal defect also increases the likelihood of degenerative changes developing in other subregions of the same tibiofemoral joint compartment [8]. Therefore, early identification of the lesion and preventative therapy are priorities [7]. However, these defects are difficult to treat due to the poor innate regenerative ability of cartilage [9]. Results from randomized controlled trials (RCTs) of surgical interventions, such as bone marrow stimulation and osteochondral transplantation, do yield radiographic and clinical improvements, although the lifespan of the repair is unclear, and it is not obvious that one surgical technique is superior to others [10,11,12].

The use of cell-based therapy for the repair of chondral defects began almost 30 years ago, involving the in vitro expansion of biopsied chondrocytes followed by implantation into the defect site and the use of a periosteal flap for coverage (ACI-P) [13]. Autologous chondrocyte implantation (ACI) has since evolved to matrix-induced ACI (MACI), a technique which uses collagen type I/III scaffolds to improve cell delivery, as well as the retention of cells within the graft [14,15]. The evidence so far demonstrates some improvement in clinical outcomes when comparing ACI or MACI to mosaicplasty and microfracture, as well as increased cost-effectiveness [16,17,18]. However, there are limitations to the conclusions that can be drawn due to the relatively short follow-up times and a small number of high-quality RCTs.

The limited capability of chondrocytes to expand in culture led, more recently, to experimentation with mesenchymal stem cells (MSCs) as a regenerative technique, which may lead to a repair cartilage phenotype more akin to native hyaline cartilage [14,19]. There is a growing interest in the use of stem-cell therapies for cartilage repair [20]. A number of human, animal and in vitro studies have been carried out using an array of synthetic polymer scaffolds, based on collagen, fibrinogen glue or other synthetic proteins. Supplementation of the scaffold with the growth factor TGF-β1 has also been tried in vitro. However, the widespread translation of basic science and animal research into clinical applications in humans is still awaited [20]. The current evidence does show superior outcomes with MSCs compared to cell-free controls in humans [21,22]. However, limited conclusions can be drawn due to the heterogeneity between studies and the small number of RCTs.

The focus of our systematic review is on the degree of integration between the repair cartilage formed from the ACI or MSC implant and the native cartilage surrounding the repaired defect. Histological evidence from in vitro studies shows that implants using cultured chondrocytes can integrate better with native cartilage as compared to cell-free controls or mosaicplasties [23,24]. The increased tensile strength at the repair site has important clinical significance. The need for the optimization of integration is further emphasized by findings that the integration boundary remains the weakest point in the repair tissue, even when visually acceptable integration is seen histologically [25]. This may have consequences for the long-term survival of an implant in vivo. Human studies are unlikely to involve histological or biomechanical analysis of the repair, but magnetic resonance imaging (MRI) is considered a high-quality, non-invasive technique which can assess the morphology of the repair, including border integration [26]. We aimed to assess the degree of integrative repair of chondral defects of the tibiofemoral joint using cell-seeded implants.

## 2. Materials and Methods

### 2.1. Search Algorithm

This review was conducted in accordance with the Preferred Reporting Items for Systematic Reviews and Meta-Analysis (PRISMA) guidelines [27]. A comprehensive literature search was conducted from conception to January 2022 using the following databases: (1) PubMed, (2) Embase, (3) MEDLINE, (4) Web of Science and (5) CINAHL. The detailed search strategy can be found in Appendix A. This review was registered in the International Prospective Register of Systematic Reviews PROSPERO (CRD42022308616).

The studies were uploaded onto the Rayyan website [28], where titles and abstracts were independently screened by EL and IEE before a subsequent full-text screen. A third (VL) and fourth reviewer (WK) were consulted for unresolvable disagreements.

### 2.2. Inclusion and Exclusion Criteria

The PICOS model (Population, Intervention, Comparison, Outcome, Study type) [29] was used to formulate the inclusion and exclusion criteria, as described in Table 1.

### 2.3. Data Extraction

The extraction of the data was independently performed by EL and IEE, with third and fourth reviewers (WK and VL) to resolve disagreements. A standardized table created in an Excel spreadsheet was populated with extracted data including:Study characteristics such as study design, cohort size and time of follow-up.Demographic information such as mean age, sex distribution, defect location and defect size.Type of intervention, including type of ACI/MSC and scaffold composition.Molecular status of the cells used in each intervention, including cluster of differentiation (CD) molecule profile and molecular construct of the scaffolds.Primary outcome measures regarding integration between the implant and native cartilage, assessment method (MRI, arthroscopy or histology) and scoring system used to quantify integration.Secondary outcomes, including clinical scores and any surgical complications.

### 2.4. Data Analysis

Regarding integration between the implant and native cartilage, outcomes from comparable MRI-based scoring systems or the proportion of patients achieving complete integration, as seen on MRI, were pooled in meta-analyses.

For studies evaluating integration using MRI, the “integration to the border zone” component was extracted from scoring systems such as the Magnetic Resonance Observation of Cartilage Repair Tissue (MOCART) score, which post-operatively assesses cartilage repair in comparison to adjacent hyaline cartilage [30,31]. The results were sometimes presented as the proportion of patients achieving a predefined MOCART border integration score: (1) complete integration with adjacent cartilage; (2) incomplete integration (split like border visible); (3) a visible demarcating border of <50% of the length of the repair tissue; or (4) a visible defect >50% of the length of the repair tissue. Other studies used the following scoring system: 1 = poor integration, 2 = fair, 3 = good, 4 = excellent, and a mean MOCART integration score was reported. The results were extracted for studies that used their own MRI composite scores, which were deemed comparable with MOCART and included in the analysis. As the MOCART scoring system is commonly used to evaluate the overall quality of the implant [30], other pertinent components were extracted such as the degree of defect filling, how intact the surface of the implant was, the homogeneity of the implant (described as its structure) and the status of the subchondral lamina and bone.

For studies that evaluated integration using second-look arthroscopy or biopsy, either the macroscopic International Cartilage Repair Society (ICRS) or histological ICRS score results were extracted.

Clinical outcomes from six scoring systems were pooled for meta-analysis: (1) the International Knee Documentation Committee (IKDC) score, a patient-reported self-evaluation on whether they can complete certain tasks; (2) the Knee Injury and Osteoarthritis Outcome (KOOS) score, a percentage score evaluating short- and long-term symptoms; (3) the Lysholm score, a score out of 100 examining the presence of specific knee symptoms such as locking and instability; (4) the Tegner Knee Activity Scale (TAS), grading the amount of work and sporting activities possible; (5) the Visual Analogue Scale (VAS), which measures pain intensity; (6) the Short Form-36 Physical and Mental scores (SF-36), short surveys regarding physical and mental health.

Meta-analyses were carried out using RStudio version 4.0.5. For continuous data, the Wan et al. estimator was used where the mean ± standard deviation was not given in the manuscript [32]. Higgins and Thompson’s I^2^ statistic and Cochran’s Q test were used as measures of heterogeneity [33,34]. Prediction intervals were also included to provide a range into which future studies’ effect sizes can be expected to fall. Subgroup analyses were performed according to whether or not collagen was a component of the implanted scaffold.

### 2.5. Assessing Risk of Bias

Risk of bias assessments were carried out independently by EL and IEE, and VL was consulted for unresolvable disagreements.

The Cochrane RoB 2.0 tool was used to assess randomized trials according to its five domains [35]: (1) bias from the randomization process; (2) bias due to deviations from the intended interventions; (3) missing outcome data; (4) bias in measurement of the outcome; (5) bias in selection of the reported result. These domains were each assessed as having a low risk, some concerns or high risk of bias, and an overall risk was determined.

The Cochrane ROBINS-I tool was used to assess non-randomized trials according to its seven domains [36]: (1) bias due to confounding variables; (2) bias in the selection of participants into the study; (3) bias in the classification of interventions; (4) bias due to deviations from intended interventions; (5) bias due to missing data; (6) bias in the measurement of outcomes; (7) bias in the selection of the reported result. These domains were assessed as having a low, moderate, serious or critical risk of bias, and an overall risk was determined.

The results of the assessments were presented using the robvis package [37] in RStudio.

## 3. Results

### 3.1. Search Results

A total of 963 papers were identified after the initial search on five databases (Figure 1). Following deduplication, 883 papers remained for title and abstract screening. A total of 124 full texts were assessed for eligibility, from which 17 studies were eligible for data synthesis.

### 3.2. Characteristics of Selected Studies

Patient demographics, study designs and interventions are displayed in Table 2. Seven included studies were RCTs, and the rest were prospective case series. Two pairs of studies reported results of the same trial at different follow-up time points [38,39,40,41]. All studies investigated the performance of ACI. One of these was a randomized trial comparing MACI with synovium-derived MSCs [42]. Three studies combined the implantation of ACIs with a bone graft for the treatment of osteochondral defects [43,44,45]. One study used a suspension of ACIs in gel composed of fibrinogen and thrombin, which was administered directly onto the chondral defect during surgery and allowed to harden [46]. The locations of the treated chondral defects were reported in 16 of the 17 included studies. Of these, all included patients with defects of the medial and/or lateral femoral condyles (MFC, LFC). Four also included patients with trochlear defects [45,46,47,48], and two included defects of the tibial plateau [46,49]. One study treated patellar defects, but reporting of individual patient data allowed for inclusion [45]. One study followed up patients for between 22 to 47 days [50]. The remaining 16 studies performed a follow-up of at least two years.

The outcomes of integration for each study are displayed in Table 3. Integration outcomes were recorded for radiographic, histological and arthroscopic data. Only two studies investigated the degree of integration using a biopsy [44,48], and another two performed second-look arthroscopy [47,51]. Other imaging outcomes are presented in Appendix A. Appendix A summarizes the performance of therapies in improving clinical scores and their surgical complications.

### 3.3. Integration Outcomes

#### 3.3.1. Magnetic Resonance Imaging

All studies, except one [47], used MRI to assess the repair cartilage formed from the implant. Of these, 11 studies used the MOCART scoring system for their evaluation. Three studies used an MRI composite score composed of the same parameters [38,39,53]. Marlovits et al. [50] and Selmi et al. [51] reported integration outcomes using their own scoring systems.

Ten studies reported the proportions of patients achieving each possible degree of integration [40,41,43,44,45,46,48,50,52,53]. Of these, seven showed that most patients had achieved completed integration with the surrounding native cartilage at the time of final follow-up, ranging from 61–100% of patients [43,44,45,46,48,50,53]. Two studies investigating the same group of patients at different time points showed that a minority of patients achieved complete integration [40,41]. Zeifang et al. showed that 38% (95% CI (9%, 76%)) of patients who received MACI and 0% (95% CI (9%, 34%)) of patients receiving ACI-P achieved complete integration at 12 months post-operatively [40]. Barié et al. demonstrated an improvement in the same group of patients, reporting that 22% (95% CI (3%, 60%)) of patients receiving MACI and 43% (95% CI (10%, 82%)) receiving ACI-P achieved complete integration at final follow-up, which was approximately 8 or more years post-operatively [41]. A meta-analysis showed that, from a total of 200 patients, 64% (95% CI (51%, 75%)) of patients achieved complete integration at endpoint (Figure 2).

Five studies (four RCTs and one case series) reported the MOCART integration score as a mean score using the following scoring system: 1 = poor integration, 2 = fair, 3 = good, 4 = excellent [38,39,42,49,54]. All showed an improvement in the mean integration score over time, although this was not always statistically significant (Table 2). A random effects meta-analysis (Figure 3) revealed a statistically significant improvement in mean integration scores post-operatively (standardized mean difference (SMD): 1.16; 95% CI (0.07, 2.24); *p* = 0.04).

Ebert et al. compared MACI and a traditional rehabilitation model (full weight-bearing at 11 weeks post-operatively) to MACI and accelerated rehabilitation (full weight-bearing at 8 weeks post-operatively) [38,39]. Both showed a significant improvement in their mean integration scores between 3 and 24 months post-operatively [38]. However, at 5-year follow-up, the integration scores had demonstrated a decline when compared to the scores at 24 months and no longer differed significantly from the first assessment at 3 months [39]. There was no significant difference between the two groups at 24 months, nor at 5 years of follow-up. Ebert et al. later performed a trial comparing a 6-week return to full weight-bearing with an 8-week return [54]. The groups did not show a statistically significant improvement in their respective integration scores over time, but the mean score of the 6-week group (3.29 ± 0.25) was significantly better compared to that of the 8-week group (2.79 ± 0.23) at 24 months post-operatively.

Akgun et al. compared the use of collagen seeded with synovium-derived MSCs with MACI [42]. The mean integration score for each group improved significantly over the course of the trial. The MSC group had consistently higher integration scores, although the difference between the groups was not statistically significant at any point.

Selmi et al. did not use a scoring system but reported that the transition zone of repair tissue with the adjacent cartilage was smooth and regular in 13 out of 15 patients and that the repair tissue could not be distinguished from the native cartilage in 11 patients [51].

#### 3.3.2. Arthroscopy

Two studies used arthroscopy to assess the quality of the repair following implantation. Selmi et al. used MACI in their prospective case series [51], obtaining a mean total ICRS score of 10 (range 5–12), which would indicate hyaline-like tissue. Out of 13 patients, 9 demonstrated either a completely integrated implant or a gap of less than 1 mm between the implant and the native cartilage. Two patients showed 75% integration of the peripheral margin of the implant and another two showed 50% integration of the margin.

Saris et al. performed an RCT comparing MACI with microfracture (MFX) [47]. They assessed the quality of the repair using the macroscopic ICRS II score. A total of 29.2% and 20.8% of patients demonstrated complete integration in the MACI and MFX groups, respectively. The difference in integration between the two groups was not statistically significant. The overall repair assessment revealed that 19.4% of the MACI group and 11.1% of the MFX group achieved a Grade I repair structure, which is defined as being “normal” cartilage.

#### 3.3.3. Histology

Two studies performed biopsies of the repair tissue post-operatively [44,48]. Bhattacharjee et al. performed a case series, using a bone graft combined with ACI as their intervention [44]. Ten biopsy specimens from eight patients were retrieved at different time-points. The specimens were stained with hematoxylin and eosin to assess the morphology of the repair. Morphological assessment was performed using the ICRS II Histology Score, for which the maximum score attainable is 10 for each domain assessed. Integration was assessed for five patients. Two patients were biopsied twice, one at 11 and 24 months post-operatively and the other at 13 and 37 months post-operatively. They demonstrated increases in integration scores from 7.85 to 9.5 and 9.8 to 10, respectively. The mean integration score at final follow-up was 9.71. Of note, nine of the ten biopsy specimens showed fibrocartilage, and one demonstrated a mixture of fibrocartilage and hyaline cartilage.

Slynarski et al. investigated the use of a copolymer of polyethylene glycol terephthalate and polybutylene terephthalate combined with ACI and mononucleated cells [48]. Histological analyses of 31 osteochondral specimens taken at a variety of time-points were conducted, ranging from six months to 24 months post-operatively. Their assessment drew upon the ICRS II and O’Driscoll grading scales. For 58.6% of patients, there was no visible interface (i.e., there was no gap) between the repair and native tissues. Cartilage was classed as hyaline-like when positively staining for collagen type II, aggrecan and sulphated glycosaminoglycans (safranin O stain), negative for collagen type I and caused no birefringency of polarized light. 71% of the specimens demonstrated hyaline-like repair, 19.4% were positive for fibrocartilage and 9.7% were composed of fibrous tissue.

### 3.4. Other Imaging Outcomes

Fifteen studies assessed the quality of cartilage repair on MRI using parameters other than integration (Appendix A). All of these reported the degree of filling of the chondral defect. Seven studies reported that a majority of patients achieved complete defect filling by final follow-up [41,43,45,46,51,52,53]. Four studies reporting mean MOCART scores demonstrated an improvement in the mean filling score over time [38,42,49,54]. Whether or not the surface of the repair was intact was also assessed by 13 studies. Five demonstrated that at least half of the enrolled patients had an intact repair surface [45,46,48,52,53]. Six patient cohorts, across a further four studies, showed mixed results, with three cohorts showing improvement over time [38,42,54], and another three showing a declining score after 3 months of follow-up [42,49,54]. An analysis of the structure of the repair tissue is also reported in 14 studies. Seven studies demonstrated that only a minority of patients achieved a homogenous repair structure [40,41,43,44,45,48,52]. Again, those studies reporting mean scores demonstrated varied results, with three patient cohorts showing improvement [38,42,49], one showing no change [42] and two showing a declining score after 3 months follow-up [38,54].

Regarding the MRI assessment of the subchondral lamina, four studies demonstrated that most patients had achieved an intact lamina at final follow-up [40,41,46,53]. Five showed the converse [43,44,45,48,52]. However, the studies reporting mean scores all demonstrated improvements in the MOCART subchondral lamina score [38,42,49,54]. The results from the assessment of the quality of the subchondral bone were also varied, with five studies demonstrating a majority with intact subchondral bone [40,45,48,51,53], and four reporting this in fewer than half of the participants [41,43,44,52]. Three studies reported an improvement in the MOCART subchondral bone score over time [42,49,54]. One demonstrated a statistically significant decline [38].

Akgun et al. performed a randomized control trial comparing MSC-seeded collagen scaffolds with MACI [42]. The MSC group performed better in all MOCART domains reported in this review, including the integration score. The difference in mean scores was statistically significant for the degree of defect filling and the surface of the implant.

Ebert et al. investigated the effect of traditional versus accelerated rehabilitation programs following MACI [38,39]. Five-year follow-up consistently demonstrated a reduction in the mean scores for each of the MOCART domains, including border integration, relative to those reported at 2 years.

Ebert and colleagues frequently measured the overall quality of the repair using an MRI or MOCART composite score [38,39,49,54]. This was derived by multiplying the score for each domain by a weighting factor and adding the scores together [49]. A significant improvement was found (standardized mean difference: 1.71; 95% CI (1.88, 3.22); *p* = 0.03), demonstrating a global improvement across the various domains of the MOCART score over the duration of follow-up (Figure 4).

### 3.5. Clinical Outcomes

Most studies reported an improvement in the clinical scores when comparing the baseline result to the same cohort of post-operative patients (Appendix A). This includes knee health indices, which improved across all studies post-operatively. Eight studies reported an improvement in IKDC score [40,41,43,46,47,48,51,52] (SMD: 1.60, 95% CI (1.24, 1.95); *p* < 0.0001) (Figure 5A). The KOOS score was reported in seven studies [42,46,47,48,49,53,54], all of which showed an improvement in the Pain (SMD: 5.58; 95% CI (3.36, 7.79); *p* = 0.0004) and Symptoms (SMD: 4.90; 95% CI (2.79, 7.02); *p* = 0.0007) KOOS subgroups (Figure 5B,C). The five studies using the Lysholm Knee Questionnaire showed a pooled improvement [40,41,43,44,49] (SMD: 2.42; 95% CI (0.31, 4.53), *p* = 0.03) (Figure 5D). The Tegner Knee Activity Scale (TAS) was reported by six studies [40,41,42,43,49,52], which showed a pooled improvement over time (SMD: 1.91; 95% CI (0.53, 3.30); *p* = 0.01) (Figure 5E).

Five studies investigated the improvement of pain symptoms using the Visual Analogue Scale (VAS) Pain Score, for which a negative score signifies a better outcome [42,45,49,53,54]. A significant improvement was observed at end point (SMD: −6.91; 95% CI (−9.92, −3.90); *p* = 0.001) (Figure 5F). Improvements in overall patient health are reflected in the SF-36 Physical and Mental health scores. The SF-36 Physical scores improved in all five studies that reported it [41,45,49,53,54] (SMD: 3.81; 95% CI (1.42, 6.19); *p* = 0.008) (Figure 5G). The SF-36 Mental health scores were reported by the same five studies, four of which demonstrated an improvement [41,45,49,53,54] (SMD: 1.52 (95% CI (−0.02, 3.06), *p* = 0.052) (Figure 5H).

### 3.6. Graft Failure and Complications

Eight of the included studies did not report any complications [38,39,40,48,49,52,53,54]. The complication most frequently reported was graft hypertrophy, which occurred in six studies at a range of frequencies (12.5% to 71% of patients) [38,40,41,49,53,54]. One study reported that a single patient required re-operation due to symptomatic hypertrophy [41]. Joint effusions occurred in three studies (14–65% of patients) [40,48,52]. Other, less commonly reported complications included post-operative adhesions [48,52], the need for re-operation [41] and persistent post-operative arthralgia [48].

Graft failures, defined as “delaminated grafts or repair sites devoid of repair tissue”, [39] were reported by eight studies, ranging from 3% to 11% [38,39,40,48,49,52,53,54]. The pooled rate of graft failure was 8% (95% CI (6%, 10%)) (Figure 6).

### 3.7. Subgroup Meta-Analyses

Subgroup meta-analyses were performed to further investigate the effect of using a collagen-based scaffold (Table 4). Out of nine studies utilizing a collagen-based scaffold, eight used a collagen type I/III scaffold [38,39,42,47,49,50,53,54]. Another used a bilayer type I collagen sponge containing chondroitin sulfate [43].

Other scaffolds were composed of an agarose-alginate hydrogel [51], a combination of polyglactin 910 and poly-p-dioxanon [40,41], a benzylic ester of hyaluronic acid (HYAFF 11) [52,55] or a polyethylene glycol terephthalate and polybutylene terephthalate copolymer [48]. One study used a gel, suspending ACIs in fibrinogen and thrombin, which was placed directly on the defect and allowed to harden during the surgical procedure [46]. Two studies used bone grafts covered by either periosteum or a collagen membrane, with ACIs injected under the covering [44,45]. These were not included in the collagen subgroup, as the ACIs were not seeded into the collagenous membrane itself before implantation and because the overall study results included the periosteal flaps.

Studies using collagen scaffolds demonstrated that 72.6% (95% CI (53.6%, 85.9%)) achieved complete integration compared to 55.9% (95% CI (38.6%, 71.9%)) for other techniques (*p* = 0.06).

The improvements in clinical scores were all superior in those using collagen scaffolds. This difference was statistically significant for the IKDC (*p* = 0.005), KOOS Pain score (*p* = 0.006), TAS (*p* = 0.009), SF-36 Physical score (*p* = 0.0004) and SF-36 Mental score (*p* < 0.0001). The proportion of patients with graft failures was also reduced for collagen scaffolds.

### 3.8. Risk of Bias Assessment

Overall, there were some concerns with the risk of bias in the randomized studies (Figure 7A). This was primarily caused by potential bias in the randomization process, because most studies did not conceal the allocation of randomized patients to each intervention arm.

For non-randomized studies, the overall risk of bias was moderate (Figure 7B). The most frequent source of potential bias was in the measurement of the outcome. This was because assessors were rarely blinded to the purposes of the intervention administered.

## 4. Discussion

In this review, we investigated the degree of integration between repair and native cartilage after using current ACI or MSC-seeded implants for focal chondral defects of the tibiofemoral joint. We selected 17 studies that contained quantitative information regarding the degree of integration, either as a score or as the proportion of patients achieving complete integration, as determined by MRI, arthroscopy or histology. Our primary findings show that ACI is associated with good quality integration. The results for clinical outcomes, including function and pain, also demonstrated improvements after the use of ACI. Although the limited evidence available suggests that MSCs can achieve improvements in integration and clinical outcomes, broad conclusions could not be drawn due to the relative lack of studies treating focal chondral defects with MSC implants. This is a common finding of other reviews, which remark that clinical studies investigating MSCs are few in number and involve small patient cohorts [56,57,58]. These and other limitations, including limited follow-up and the heterogeneity of data, are evaluated in this discussion.

### 4.1. MRI as an Investigative Technique for Integration

The results of the meta-analyses for integration were encouraging, with 64% of 200 patients achieving complete integration (95% CI (51%, 75%)). Only in two of the ten comparable papers did a minority of patients undergoing ACI achieve complete integration [40,41]. These were studies investigating the same group of patients and interventions at different timepoints.

MRI was by far the most popular choice of investigation, with almost all (16/17) using it to evaluate repair cartilage morphology. This is likely because MRI is widely available, non-invasive and validated as a technique for assessing repair cartilage [26]. Scanners of similar specifications (high-resolution MRI at least 1.0 to 3.0 T) and sequencing were used across the studies. All looked at the tibiofemoral joint from both coronal and sagittal planes, with the fast-spin echo (dual T2-FSE) and fat-suppressed gradient echo sequences (3D-GE-FS), in accordance with the approach used by Marlovits et al. [50]. Although data was available for most patients, there were concerns over the potential risk of bias due to missing outcome data for both RCTs and non-randomized studies. Most RCTs reported using at least one radiologist, blinded to the patients’ clinical details and to the procedure. Blinding to the procedure usually did not occur in the non-randomized case series, possibly introducing reporting bias.

The studies did not report integration outcomes consistently, limiting the amount of comparable data available. Ten studies reported the proportion of patients achieving complete integration [31,40,41,43,44,45,46,48,52,53], and five reported the mean border zone integration score [38,39,42,49,54]. As a result, not all quantitative results were comparable, limiting the number of patients included in each analysis. Consistent reporting of the integration score would allow for comparison between a greater number of studies and increase the power of any subsequent meta-analysis.

Our review highlighted the importance for further studies to record integration specifically. Using an overall or composite MOCART score without reference to individual domains did not allow for interpretation of the degree of integration in many of the studies eligible for full-text screening. The evidence already suggests that MOCART scores might not correlate well with patient characteristics and surgical outcomes [30]. Therefore, if we are to understand how integration relates to improved clinical outcomes, future studies will have to consider integration independently.

The significant statistical heterogeneity (I^2^ = 90%) revealed by our analysis of the comparable MOCART border integration scores means that a cautious interpretation of the statistically significant improvement (SMD: 1.16; 95% CI (0.07, 2.24; *p* = 0.04) should be made. Variance in the pooled results of multiple studies is often due to a random sampling error or clinical heterogeneity [59]. The patient baseline characteristics were similar for the studies included in the meta-analysis, so it is unlikely that the errors arose from non-random sampling. Clinical heterogeneity, on the other hand, is more likely to have contributed to this. This might be the result of factors such as (1) differences in treatment, (2) differences in study design or (3) differences in data analysis methods [59]. In our meta-analysis, the four studies performed by Ebert et al. used a similar intervention, MACI, but involved different rehabilitation protocols and follow-up times. The statistical heterogeneity observed here may be partially attributed to these differences. Furthermore, Akgun et al. used MACI in one patient cohort and MSC-seeded collagen scaffolds in the other [42]. This difference in interventions may have contributed to statistical heterogeneity, possibly making the pooled effect size unreliable.

We focused on integration because of its perceived clinical relevance to the durability of a repair. However, it is important to recognize that there are additional parameters used to assess for the quality of cartilage repair. Interestingly, in two papers investigating the same patients over time, a minority achieved complete integration [40,41], but “defect filling” was achieved in most patients at end point. For MACI and ACI-P, the percentage of patients achieving complete defect filling was 50.0% and 11.1%, respectively, at 24 months. These values had increased to 55.5% and 71.4% by the time of final follow-up, which was performed at 9.6 ± 0.9 years (MACI) or 8.6 ± 0.8 years (ACI-P). One may interpret these results as a long-lasting success in the repair of a cartilage defect, even though integration did not improve. As we have done for integration, defect fill and other MOCART domains warrant formal investigation.

### 4.2. Molecular Analysis for Optimal Scaffold and Source of Cells

A major factor contributing to clinical heterogeneity was variation in the making of implants, primarily regarding cell retrieval and culture and the choice of scaffold composition. Not every study reported the location from which cells were retrieved. Eleven studies mentioned non-weightbearing zones as the source, and fewer mentioned specifically that the lateral or medial femoral condyle or intercondylar notch was the location biopsied. The medium with which cells were cultured was often omitted, but, in those that did mention it, media included serum of the patients’ own blood and Ham’s F12 containing 10% fetal calf serum [42]. Culture duration ranged from 3 days [40] to 8 weeks [39]. Many studies did not report the cell density in scaffolds, but, among those that did, there was a wide range from 2 to 30 million cells/scaffold.

The ACI technique, including the scaffold type, varied among the studies. MACI was the most common. Two studies had used ACI-P, the more traditional and now less clinically relevant technique, albeit as a control [40,41], and a single study used gel-type ACI (GACI) [46]. The most common scaffold used was a type I/III collagen scaffold. Others included an agarose-alginate hydrogel [51], a bilayer type I collagen sponge containing chondroitin sulfate (Novocart 3D) [43], a benzylic ester of hyaluronic acid (Hyalograft C) [52,55], a polyglactin 910 combined with poly-p-dioxanon [40,41] and a polyethylene glycol terephthalate and polybutylene terephthalate copolymer [48]. As demonstrated by our subgroup analysis, collagen scaffolds were associated with improved integration, clinical outcomes and a lower graft failure rate. It is difficult to find further evidence to corroborate this in the current literature. One systematic review found only weak evidence of superiority of MACI relative to ACI-P [60]. A prospective series comparing failure rates of different ACI techniques found that altered polymer combinations with collagen performed differently [61]. Their ACI-seeded fibrin-collagen patch demonstrated fewer failures than the collagen-hydroxyapatite scaffold and alginate-agarose hydrogel scaffold.

Variations in the therapeutic process are well documented in the literature, with other reviews sharing the same observation. In a systematic review of various ACI studies, Migliorini et al. suggested that a lack of consensus on what is the best combination of cell type and method for producing implants, as well as continuous innovation and novel scaffold types, have contributed to a wide variety of available ACI therapies [62]. Establishing the most effective combination of these factors will be essential if outcomes after ACI therapy are to be optimized. This applies not only to integration but also to other morphological and clinical outcomes.

Evaluation of the molecular profile of cells has already been used to confirm the presence of MSCs or autologous chondrocytes before seeding them into scaffolds. Akgun et al. used flow cytometry or PCR to test for the expression, by chondrocytes, of CD44 and CD73 and the lack of CD45, as a quality assessment, since this expression profile is associated with a better differentiation capacity [42]. Research has shown that other markers might also be relevant, such as S-100, aggrecan TGF-β, glucocorticoid receptor alpha and the vitamin D3 receptor, which are associated with low rates of apoptosis [63]. With no consensus on what cell type to use, investigating the molecular profile of autologous chondrocytes may be a useful screening tool to determine which profile improves integration. This characterization is likely to be even more important for MSCs, given the wider range of tissues from which MSCs can be cultivated, including adipose, peripheral blood and bone marrow. In fact, Park et al. found that a substantial number of errors have been made in labelling the therapeutic MSCs used for cartilage repair [64], making an objective screening method such as the molecular profile even more pertinent.

MSCs remain a promising yet under-investigated therapeutic option for patients with focal chondral defects. Migliorini et al. showed in their review that MSCs resulted in significant improvements post-operatively, including in clinical scores such as the KOOS [62]. Another benefit of using MSCs is that it avoids the necessity of primary arthroscopic cartilage harvesting for ACI. Akgun et al. found that, for some assessments, the use of MSCs outperformed the MACI technique [42]. Given these positive but limited results, further investigation is certainly warranted for the use of MSCs in treating chondral defects, including the need to optimize MSC implant integration.

Currently, widespread uptake of MSC implant therapies in clinical practice has not yet taken place, with most data being derived from pre-clinical studies [57,58]. As well as the relatively small number of human studies, there are limited follow-up data available in comparison to ACI [57,58,62]. This not only limits our knowledge of the long-term efficacy of MSC implants but also has implications for potential safety issues. The possibility of tumors developing from implanted MSCs and concerns over differentiation into unwanted tissue types have been raised [57]. However, these have not yet been realized clinically in the investigation of cartilage repair. A greater number of human studies with continuous follow-up would provide valuable long-term data regarding the safety and efficacy of this therapy.

There are also several limitations to our current understanding, of which MSC delivery protocol is the most effective. This is partially due to heterogeneity in the therapeutic compositions used to date, recognized by others as an obstacle to collating data in meta-analyses [56]. Uncertainty regarding the best cell source, cell dosage, the presence of growth factors and rehabilitation protocols all contribute to heterogeneity [57,58]. Investigated techniques reported in the literature demonstrate wide variability in each of these domains [20], limiting the amount of comparison which can be made between studies. This is compounded by the small number of randomized comparative studies in humans, with case series of few patients predominantly making up the evidence base [56,57]. A greater number of randomized, double-blinded control trials to investigate the efficacy of MSC implants relative to established therapies, such as MACI, would contribute towards less biased evidence. Knowledge of the best treatment protocol would also be furthered by clinical studies comparing different MSC therapies, with the aim of eliciting the best cell source, dose and supplementary growth factors [57]. Advancements in each of these domains have led to agreement on a technique, MACI, which is now widely used for chondrocyte implantation. This has, to some extent, mitigated the heterogeneity between studies that had previously challenged teams implementing ACI [57,65]. The hope is that the same could be achieved for MSC implant therapies to homogenize treatment protocols and make data more comparable.

### 4.3. Role of Arthroscopy and Histology

Selmi et al. found that a majority of patients achieved complete integration, based on the macroscopic ICRS scoring, while Saris found that only a minority of patients achieved this after MACI. With only two studies available, a meta-analysis could not be performed. While Saris et al. assessed many subjects arthroscopically (60 patients), Selmi et al. assessed only a small cohort of 13 subjects. This appears to be a common theme among studies and is likely due to patients disliking the invasive nature of arthroscopy compared to MRI and the ethical issue of subjecting a patient who is already satisfied with their improved knee symptoms to another procedure [47,51].

Biopsy remains the “gold-standard” for determining integration of the repair cartilage, since MRI is less accurate and results in more inter-observer variability [66]. Two studies demonstrated good quality integration on a histological assessment [44,48]. Furthermore, biopsy allows for the determination of the repair cartilage phenotype, which varied in the included studies. Hyaline cartilage, like that of the native knee cartilage, is more desirable than fibrocartilage, which demonstrates poorer mechanical properties [48]. Biopsies at the border between implant and native tissue would certainly give a better idea of integration, demonstrated by the homogeneity of hyaline cartilage across the implant-native cartilage interface. Immunohistochemical analysis would be a sensitive method for detecting this homogeneity, as well as the cartilage phenotype, by indicating a tissue that is either made predominantly of type II collagen (hyaline cartilage) or type I collagen and IIA procollagen (fibrocartilage) [67].

### 4.4. Long-Term Outcomes

Sixteen selected studies included follow-up to at least 24 months, but only eight performed additional measurements up to five years, and three studies included follow-up beyond that. This meant that it was difficult to make conclusions about long-term outcomes. In our review, graft failure was seen in a small proportion of patients, but, without long-term measurements, it is impossible to determine whether this changes over time. This seems to be common in the current literature. Mistry et al. describe how the lack of long-term follow-up and the simultaneous quick evolution of ACI means that long-term data is for outdated techniques [16]. Understanding how integration and other outcomes change in the long-term is essential to providing a current indication of performance to be improved upon and determining which additional complications might occur, such as the need for re-operation. Prioritizing this investigation seems reasonable, as some studies have already demonstrated deteriorating outcomes over time. For example, Ebert and colleagues found that integration, and other MOCART outcomes, were poorer at five years compared to 24 months post-operatively [38,39]. Ochs et al. found that the Lysholm and IKDC scores either plateaued or decreased from 36 to 48 months [43]. Given that the nature of change in both imaging and clinical outcomes over time is not fully understood, we suggest that more authors follow-up their patients in the longer-term to elucidate this.

### 4.5. Strengths and Limitations

Our review possesses several strengths that enable us to make meaningful conclusions. The search strategy was extensive, including any study that investigated integration, and has allowed us to thoroughly extract data regarding multiple outcome measures, including MRI, arthroscopic and histological assessments, as well as clinical data. Robust inclusion and exclusion criteria allowed for a valid comparison of included studies. By extracting quantitative data, we were able to conduct meta-analyses to give more precise interpretations across the pooled studies.

However, there are limitations to the included studies. They show a high degree of heterogeneity in the scaffold composition and the length of follow-up. Ten of the seventeen studies are non-blinded, non-randomized case series, which have demonstrated a moderate risk of bias, particularly due to missing data and in the measurement of outcomes related to repair cartilage morphology. Common across many papers was a small sample size, meaning they might have been underpowered.

The largest limitation was that few analyses could be performed for MSC implants, as only one paper investigating MSCs met the inclusion criteria. Common reasons for the exclusion of MSC studies were the enrolment of patients with diffuse osteoarthritis, the use of intra-articular injections rather than implants and indistinguishable reporting of the results of treating patellofemoral and tibiofemoral joints. The inclusion of these papers would have resulted in an invalid comparison of patients with different defect types, cell-delivery methods and defect locations, respectively.

## 5. Conclusions

This systematic review and meta-analysis has collated integration outcomes following the treatment of focal chondral defects of the tibiofemoral joint with cell-seeded scaffolds. Though there were insufficient papers to make generalizations about MSC therapies, the studies we have selected suggest that the degree of integration between chondrocyte-seeded scaffolds and native cartilage appears to be of good quality in most patients. This is associated with simultaneous improvements in clinical scores. More evidence for the integration of MSC-based implants is awaited, but the current results are encouraging and suggest that MSCs may be superior to ACI in achieving an integrated repair structure. There is heterogeneity in cell sources, scaffolds and the processing of cultured cells between studies. We suggest that molecular methods, such as the cluster of differentiation (CD) characterization, should be used to screen for the quality of implanted cells and that the widespread use of collagen scaffolds should be adopted. In addition, more consistent recording of integration would allow for greater comparison between studies. By conducting this review, we hope to have established a baseline standard to which further investigations can be compared to optimize integrative repair and tackle the significant consequences of chondral defects.

## Figures and Tables

**Figure 1 ijms-23-04065-f001:**
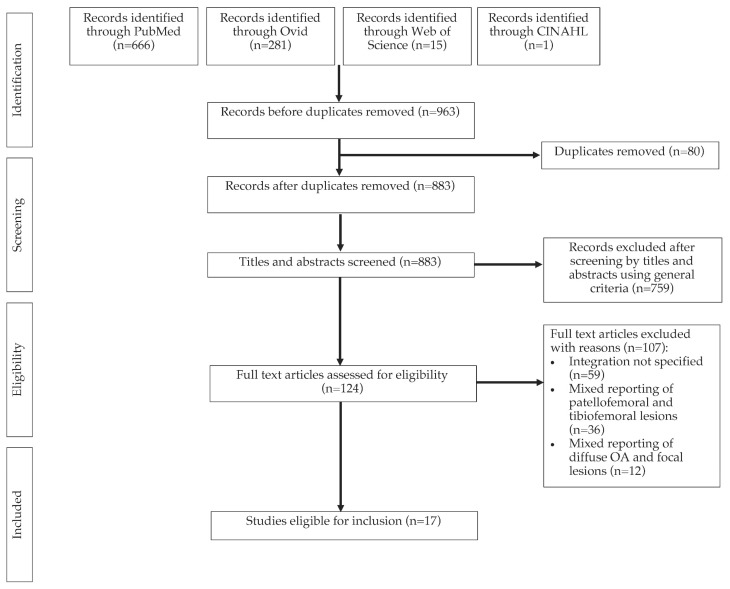
A PRISMA diagram of the study selection process.

**Figure 2 ijms-23-04065-f002:**
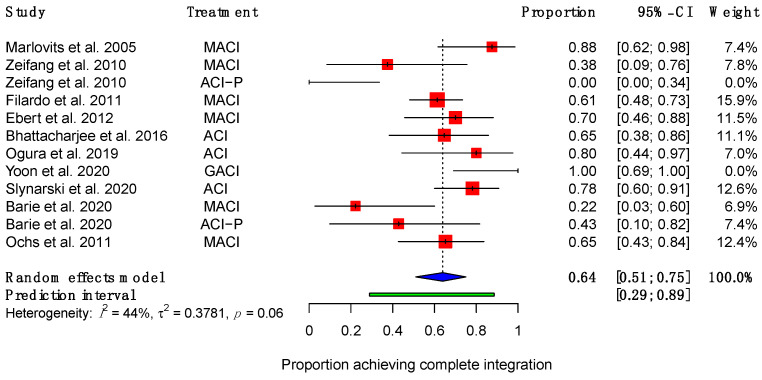
Forest plot on the proportion of patients achieving complete integration after receiving ACI therapy. (Abbreviations: MACI, Matrix-induced autologous chondrocyte implantation; ACI-P, ACI with periosteal flap cover; GACI, gel-type ACI; CI, Confidence Intervals).

**Figure 3 ijms-23-04065-f003:**
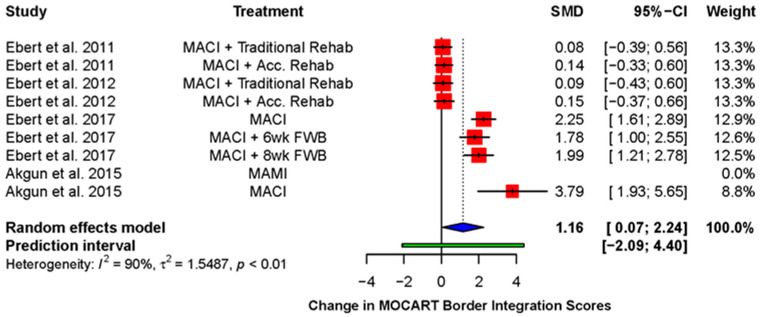
Forest plot on the improvement in mean integration score during follow-up after receiving therapy with a cell-seeded scaffold. (Abbreviations: MACI, Matrix-induced autologous chondrocyte implantation; MAMI, Matrix-induced autologous mesenchymal stem cell implantation; Acc., accelerated; FWB, full weight-bearing; SMD, standardized mean difference; CI, Confidence Intervals).

**Figure 4 ijms-23-04065-f004:**
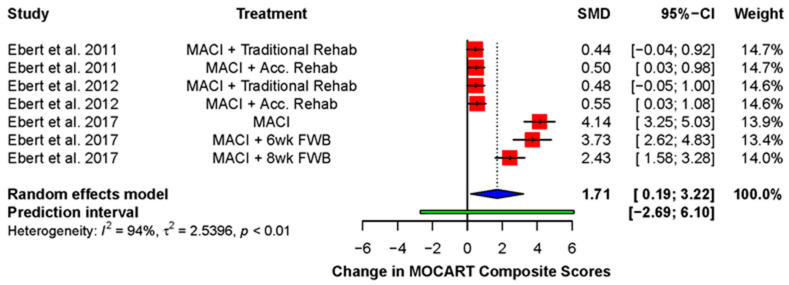
Forest plot on the improvement in MOCART composite score during post-operative follow-up. (Abbreviations: MACI, Matrix-induced autologous chondrocyte implantation; Acc., accelerated; FWB, full weight-bearing; SMD, standardized mean difference; CI, Confidence Intervals).

**Figure 5 ijms-23-04065-f005:**
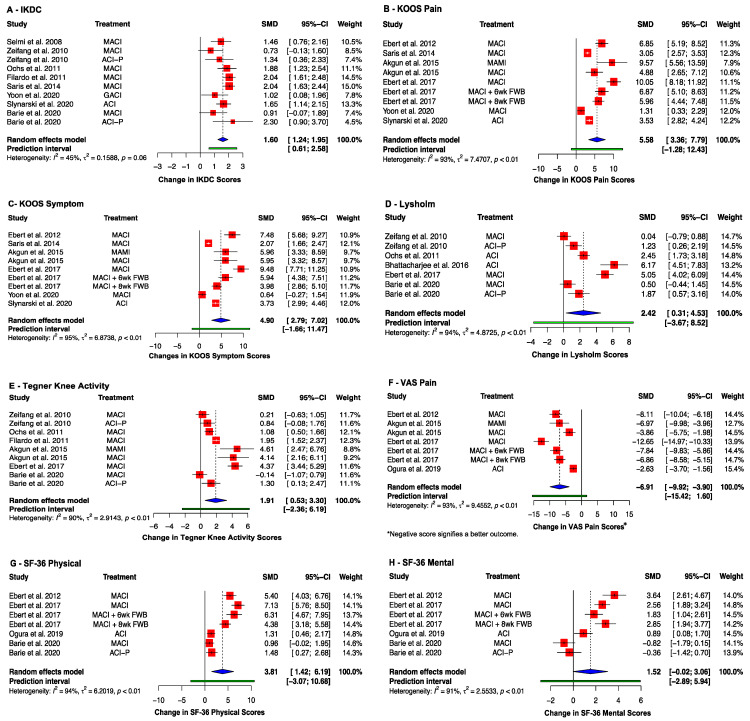
Forrest plots on the improvement of clinical scoring systems after receiving implant therapy: (**A**) IKDC; (**B**) KOOS (Pain); (**C**) KOOS (Symptom); (**D**) Lysholm; (**E**) Tegner Knee Activity Scale; (**F**) VAS Pain; (**G**) SF-36 Physical; (**H**) SF-36 Mental. (Abbreviations: MACI, Matrix-induced autologous chondrocyte implantation; ACI-P, ACI with periosteal flap cover; GACI, gel-type ACI; MAMI, Matrix-induced autologous mesenchymal stem cell implantation; FWB, full weight-bearing; IKDC, International Knee Documentation Committee; KOOS, Knee injury and Osteoarthritis Outcome Score; SF-36, Short Form-36; VAS, Visual Analogue Scale; SMD, standardized mean difference; CI, Confidence Intervals).

**Figure 6 ijms-23-04065-f006:**
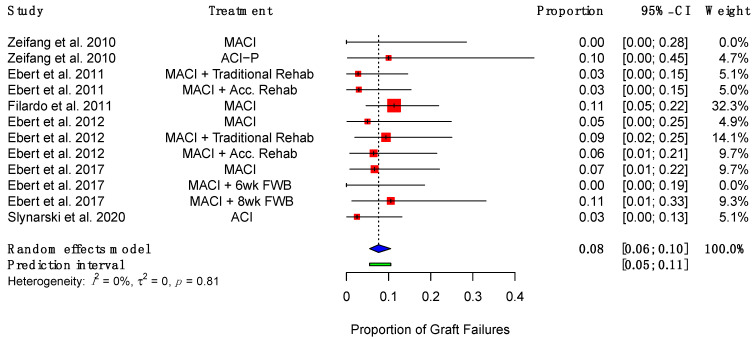
Forest plot on the proportion of implanted grafts which failed. (Abbreviations: MACI, Matrix-induced autologous chondrocyte implantation; ACI-P, ACI with periosteal flap cover; Acc., accelerated; FWB, full weight-bearing; CI, Confidence Intervals).

**Figure 7 ijms-23-04065-f007:**
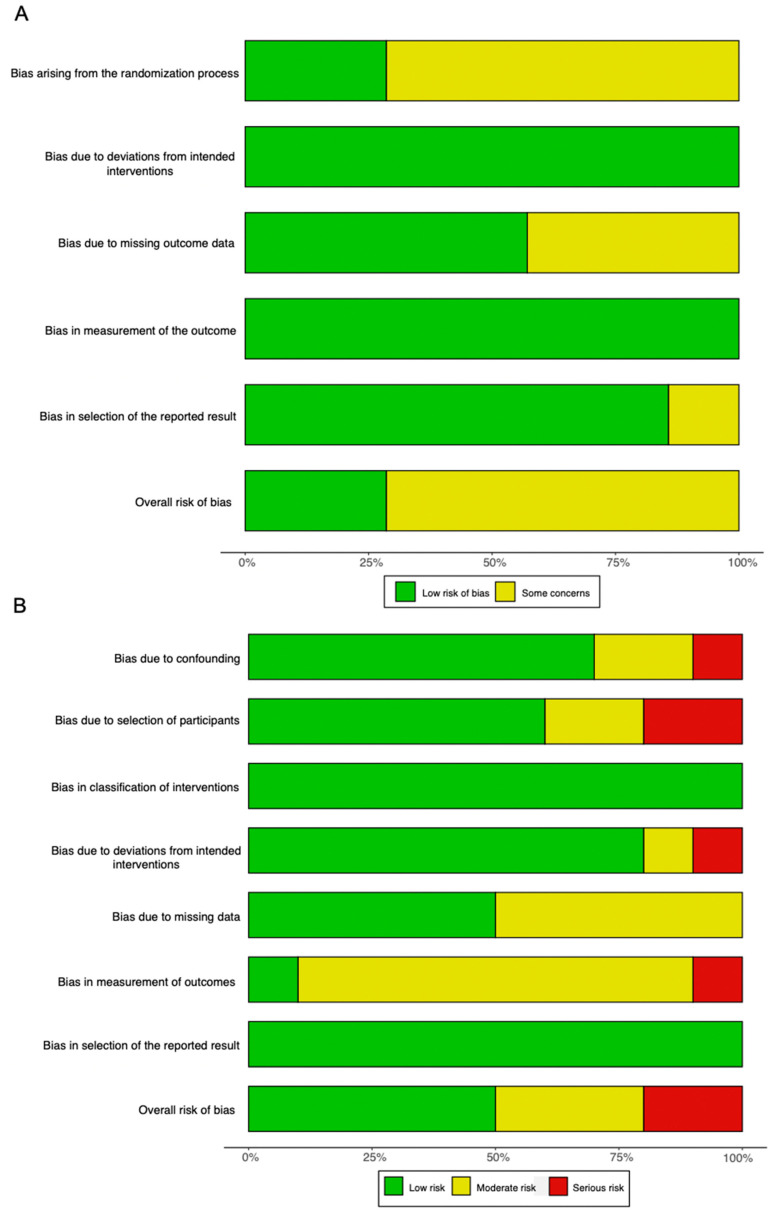
Summary graphs showing the overall risk of bias analysis using (**A**) the RoB 2.0 tool in randomized studies and (**B**) the ROBINS-I tool in non-randomized studies.

**Table 1 ijms-23-04065-t001:** PICOS inclusion and exclusion criteria for study selection.

Domain	Inclusion Criteria	Exclusion Criteria
**Population**	Studies with human patients of any age, gender and ethnicity who have focal chondral defects of the tibiofemoral joint.	Studies with patients who have diffuse osteoarthritis. Studies conducted on animals or cadavers. Ex vivo, in vitro and in silico studies.
**Intervention**	Studies that use implants made with autologous chondrocytes or mesenchymal stem cells. Studies using cell-seeded scaffolds.	Studies that do not use implants, intra-articular injection or cell-free scaffolds.
**Comparison**	Studies that compare use of implant to cell-free therapies such as microfracture.	None
**Outcome**	Studies that describe the integration between the novel and native cartilage.	Studies where the outcomes of tibiofemoral and patellofemoral defects cannot be differentiated. Studies that do not have a measuring or scoring system for integration. Studies that do not report integration separately.
**Study Type**	English with full text available. Sample size greater than 10 patients.	Case report, case series with fewer than 10 patients and review article.

**Table 2 ijms-23-04065-t002:** Study design and intervention details.

Author	Study Design	Cohort Size	Mean Age (Years)	Sex (% Male)	Defect Location	Mean Defect Surface Area (cm^2^) or Diameter (cm)	Intervention	Scaffold Composition	Cell Number/Density in Scaffold	Follow-Up
Marlovits et al., 2005 [50]	Prospective case series	16	33.1 (range 20.1–44.3)	93.8	MFC: 10/16, LFC: 6/16	4.70 ± 2.30	MACI	Type I/III collagen membrane	15–20 million cells	Mean 34.7 days post-operation, range 22–47 days
Selmi et al., 2008 [51]	Prospective case series, multi-center	20	30.0 (range 17.0–42.0)	71.5	Not Specified	3.00 (range 1.00–5.10)	MACI	Agarose-alginate hydrogel scaffold (CARTIPATCH; Tissue Bank of France, Lyon, France)	10 million cells/mL of hydrogel	3, 6, 12, 24 months
Zeifang et al., 2010 [40]	RCT	11	29.1 ± 7.5	54.5	MFC: 18/21, LFC: 3/21	4.30 ± 1.1	MACI	Polyglactin 910 and poly-p-dioxanon	20 million cells	3, 6, 12, 24 months
		10	29.5 ± 11.1	100		4.10 ± 0.90	ACI-P	N/A	15 million chondrocytes/500 μL suspension	
Ebert et al., 2011 [38]	RCT	35	39.8 (range 16.0–63.0)	62.9	MFC: 26/35, LFC: 9/35	3.31 (range 0.75–10.0)	MACI & Traditional rehabilitation	Type I/III collagen membrane (ACI-Maix Matricel GmbH, Germany)	Not Reported	3, 12, 24 months
		34	36.3 (range 21.0–62.0)	64.7	MFC: 26/34, LFC: 8/34	3.22 (range 0.65–10.0)	MACI & Accelerated rehabilitation			
Ochs et al., 2011 [43]	Prospective case series	26	Not Reported	69.2	MFC: 22/26, LFC: 4/26	5.30 ± 2.30	MACI	Novocart 3D, a bilayer collagen type I sponge with chondroitin sulfate and a bone graft (TETEC AG, Reutlingen, Germany):	Not Reported	39.8 ± 12.0 months
Filardo et al., 2011 [52]	Prospective case series	62	28.1 ± 11.4	77.4	MFC: 45/62, LFC: 17/62	2.50 ± 1.00	MACI	Hyalograft C (Fidia Advanced Biopolymers Laboratories, Padova, Italy), a benzylic ester of hyaluronic acid (HYAFF 11)	Not Reported	1, 2, 3, 4, 5, 6, 7 years
Ebert et al., 2012 [53]	Prospective case series	20	34.0 (range 16.0–57.0)	50.0	MFC: 11/20, LFC: 3/20, MTP: 2/20, LTP: 4/20	2.72 (range 1.00–5.00)	MACI	Type I/III collagen membrane (ACI-Maix; Matricel GmbH, Herzogenrath, Germany)	Not Reported	3, 12, 24 months
Ebert et al., 2012 [39]	RCT	32	39.8 (range 16.0 to 63.0)	62.9	MFC: 26/32, LFC: 9/32	3.31 (range 0.75–10.0)	MACI & Traditional rehabilitation	Type I/III collagen membrane (ACI-Maix Matricel GmbH, Germany)	Not Reported	5 years
		31	36.6 (range 21.0–62.0)	64.7	MFC: 26/31, LFC: 8/31	3.22 (range 0.65–10.0)	MACI & Traditional rehabilitation			
Saris et al., 2014 [47]	RCT	72	34.8 ± 9.8	62.5	MFC: 54/72, LFC: 13/72, T: 5/72	34.8 ± 9.20	MACI	Porcine-derived collagen type I/III membrane	500,000–1 million cells/cm^2^	24 months
		72	32.9 ± 8.8	66.7	MFC: 53/72, LFC: 15/72, T: 4/72	32.9 ± 8.80	Microfracture	N/A	N/A	
Akgun et al., 2015 [42]	RCT	7	32.3 ± 7.9	57.1	MFC: 5, LFC: 2	2.90 ± 0.80	MAMI: Synovium-derived MSCs, CD105+, CD73+, CD90+	Type I/III- collagen (Chondro-Gide^®^; Geitschlich Biomaterials)	Not Reported	
		7	32.7 ± 10.4	57.1	MFC: 5, LFC: 2	3.00 ± 0.80	MACI: Cartilage-derived chondrocytes, CD44+, CD73+	Type I/III collagen (Chondro-Gide^®^; Geitschlich Biomaterials)	Not Reported	3, 6, 12, 24 months
Bhattacharjee et al., 2016 [44]	Prospective case series	17	27.0 ± 7.0	Not Reported	MFC: 15/17, LFC: 2/17	4.50 ± 2.60	ACI, OsPlug technique	Autologous bone graft and ACI in serum injected under a periosteal or collagen cover	Not Reported	1 and 5 years
Ebert et al., 2017 [49]	Prospective case series	31	35.3 (range 16.0–57.0)	48.4	MFC: 18/31, LFC: 7/31, MTP: 2/31, LTP: 4/31	2.52 (range: 1.00–5.00)	MACI	Type I/III collagen membrane (ACI-Maix Matricel GmbH, Germany)	Not Reported	3 and 6 months, 1, 2, and 5 years
Ebert et al., 2017 [54]	RCT	18	36.4 (range 21.0–55.0)	50.0	MFC: 13/18, LFC: 5/18	3.15 (range: 1.00–6.25)	MACI and 6-week return to full weight-bearing	Type I/III collagen membrane (ACI-Maix Matricel GmbH, Germany)	Not Reported	4 and 8 weeks, 3, 6, 12, and 24 months
		19	36.4 (range 23.0–53.0)	63.0	MFC: 14/19, LFC: 5/19	2.89 (range, 1.00–7.70)	MACI and 8-week return to full weight-bearing	Type I/III collagen membrane (ACI-Maix Matricel GmbH, Germany)	Not Reported	
Ogura et al., 2019 [45]	Prospective case series	13	26.0 (range 16.0–42.0)	76.9	MFC: 8/13, T: 5/13	6.40 (range, 1.50–13.5)	ACI, segmental sandwich technique	Bone graft, periosteal patch glued with Tisseel fibrin glue (Baxter BioSurgery): 5/13, or porcine type I/III bilayer collagen membrane (Bio-Gide; Geistlich Pharma): 8/13, and ACI injected between two membrane layers	Not Reported	Mean: 7.8 ± 3.0 years, range: 2–15 years
Yoon et al., 2020 [46]	Prospective case series	10	40.3 ± 10.3	50.0	MFC: 6/10, T: 3/10, LTP: 1/10	2.90 ± 1.20	GACI	Gel composed of 1 ml fibrinogen and 0.1–0.2 ml thrombin	24–30 million cells/2 mL	3, 6, 12 and 24. 5 years for clinical outcomes
Słynarski et al., 2020 [48]	Prospective case series	40	35.2 (range 20.0–53.0)	70.0	MFC: 24/40, LFC 9/40, T: 5/40, MFC/T: 2/40	2.09 (range 1.00–3.24)	ACI combined with bone marrow mononucleated cells	Polyethylene glycol terephthalate and polybutylene terephthalate copolymer	30 million cells/cm^3^	3, 6, 12, 18 and 24 months
Barié et al., 2020 [41]	RCT	9	30.4 ± 6.8	44.4	MFC: 8/9, LFC: 1/9	4.27 ± 0.20	MACI	Fibrin combined with polyglactin 910 and poly-p-dioxanon scaffold	20 million cells/scaffold	12 and 24 months and longest follow-up (mean: 9.6 ± 0.9 years)
		7	28.8 ± 9.1	100.0	MFC: 7/7	4.08 ± 0.44	ACI-P	N/A	20 million cells/500μL suspension	12 and 24 months and longest follow-up (mean: 8.6 ± 0.8 years)

Abbreviations: RCT, randomized control trial; MFC, medial femoral condyle; LFC, lateral femoral condyle; MTP, medial tibial plateau; LTP, lateral tibial plateau; T, trochlea; MACI, matrix-induced autologous chondrocyte implantation; ACI-P, ACI with periosteal flap cover; GACI, gel-type ACI; MAMI, Matrix-induced autologous mesenchymal stem cell implantation; MSC, mesenchymal stem cell; CD, cluster of differentiation.

**Table 3 ijms-23-04065-t003:** MRI, Arthroscopic and Histological assessments of integration.

Author	Number of Participants	Scoring System	Results	*p*
Marlovits et al., 2005 [50]	16	MRI evaluation of cartilage interface	Completely attached: 14/16 (87.5%), partially attached: 1/16 (6.25%), detached: 1/16 (6.25%)	Not assessed
Selmi et al., 2008 [51]	15	MRI evaluation of cartilage interface	Transition zone of repair tissue with adjacent normal cartilage smooth: 13/15 (86.7%), repair tissue no longer distinguished from adjacent normal cartilage: 11/15 (73.3%)	Not assessed
	13	Arthroscopy: macroscopic ICRS assessment	Complete integration: 9/13 (69.2%), 75% of peripheral margin integrated: 2/13 (15.4%), 50% of peripheral margin integrated: 2/13 (14.4%)	Not assessed
Zeifang et al., 2010 [40]	8 (MACI)	Overall mean MOCART, lower score corresponds to more normal MRI	6 months: 7.0 ± 2.7, 12 months: 6.3 ± 3.5, 24 months (*n* = 7): 6.3± 3.0	(m-ACI vs ACI-P) 6 months: *p* = 0.0123, 12 months: *p* = 0.02065, 24 months: *p* = 0.6926
		MOCART border zone integration	Complete integration at 6 months: 2/8 (25.0%), complete integration at 12 months: 3/8 (37.5%)	Not assessed
	9 (ACI-P)	Overall mean MOCART, lower score corresponds to more normal MRI	6 months: 10.3 ± 1.6, 12 months: 8.4 ± 2.2, 24 months (*n* = 11): 6.8 ± 4.7	(m-ACI vs ACI-P) 6 months: *p* = 0.0123, 12 months: *p* = 0.02065, 24 months: *p* = 0.6926
		MOCART border zone integration	Complete integration at 6 months: 0/9 (0%), complete integration at 12 months: 0/9 (0%)	Not assessed
Ebert et al., 2011 [38]	34 (MACI & traditional rehabilitation)	Mean MRI composite	3 months: 2.80 (SE = 0.10), 12 months: 3.15 (SE = 0.12), 24 months: 3.07 (SE = 0.11)	Time effect *p* < 0.0001, group effect *p* = 0.740, interaction effect = 0.796
		Mean border zone integration	3 months: 2.66 (SE = 0.18), 12 months: 2.84 (SE = 0.19), 24 months: 2.75 (SE = 0.20)	Time effect *p* = 0.0113, group effect *p* = 0.659, interaction effect *p* = 0.733
		Percentage of patients achieving good to excellent for border integration	3 months: 68%, 12 months: 71%, 24 months: 68%	Not assessed
	35 (MACI & accelerated rehabilitation)	Mean MRI composite	3 months: 2.81 (SE = 0.10), 12 months: 3.21 (SE = 0.13), 24 months: 3.14 (SE = 0.12)	Time effect *p* < 0.0001, group effect *p* = 0.740, interaction effect = 0.796
		Mean border zone integration	3 months: 2.77 (SE = 0.19), 12 months: 2.90 (SE = 0.19), 24 months: 2.93 (SE = 0.21)	Time effect *p* = 0.0113, group effect *p* = 0.659, interaction effect *p* = 0.733
		Percentage of patients achieving good to excellent for border integration	3 months: 68%, 12 months: 71%, 24 months: 76%	Not assessed
Ochs et al., 2011 [43]	23	Overall mean MOCART	At latest follow-up: 62.4 ± 18.9	3 months compared to all other follow-up points: 0.008 ≤ *p* < 0.001
		MOCART border zone integration	Complete integration: 15/23 (65.2%), incomplete integration: 5/23 (21.7%), visible defect: 3 (13.0%)	Not assessed
Filardo et al., 2011 [52]	42	MOCART border zone integration	Complete integration: 26/42 (62.0%)	Not assessed
Ebert et al., 2012 [53]	20	MRI composite (proportion rated good to excellent)	3 months: 14/20 (70.0%), 12 months: 16/20 (80.0%), 24 months: 14/20 (70.0%)	Not assessed
		MRI composite (proportion rated poor to fair)	3 months: 6/20 (30.0%), 12 months: 4/20 (20.0%), 24 months: 6/20 (30.0%)	
		Border zone integration (proportion rated good to excellent)	3 months: 12/20 (60.0%), 12 months: 14/20 (70.0%), 24 months: 14/20 (70.0%)	Not assessed
		Border zone integration (proportion rated poor to fair)	3 months: 8/20 (40.0%), 12 months: 7/20 (35.0%), 24 months: 6/20 (30.0%)	
Ebert et al., 2012 [39]	29 (MACI & traditional rehabilitation)	Mean MRI composite	2.91 (SE = 0.17)	Mean MRI composite: traditional vs accelerated *p* = 0.614, Border integration: traditional vs accelerated *p* = 0.138
		Mean border zone integration	2.50 (SE = 0.22)	
	29 (MACI & accelerated rehabilitation)	Mean MRI composite	3.01 (SE = 0.12)	
		Mean border zone integration	2.92 (SE = 0.17)	
Saris et al., 2014 [47]	60 (MACI)	Arthroscopy: macroscopic ICRS assessment	Complete integration: 45/60 (62.5%), demarcating border < 1 mm: 10/60 (13.9%), ¾ integrated or ¼ with border > 1 mm: 4/60 (5.60%), ½ integration and ½ with border > 1 mm: 4/60 (5.60%), no contact to ¼ integrated: 1/60 (1.40%), missing: 8/60 (11.1%)	MACI vs Microfracture *p* = 0.519
	56 (Microfracture)	Arthroscopy: macroscopic ICRS assessment	Complete integration: 38/60 (52.8%), demarcating border < 1 mm: 9/60 (12.5%), ¾ integrated or ¼ with border > 1 mm: 7/60 (9.70%), ½ integration and ½ with border > 1 mm: 3/60 (4.20%), no contact to ¼ integrated: 2/60 (2.80%), missing: 13/60 (18.1%)	
Akgun et al., 2015 [42]	7 (MSCs)	Mean MOCART border zone integration	3 months: 2.29 ± 0.49, 12 months: 3.00 ± 0.00, 24 months: 3.57 ± 0.53	Time effect *p* = 0.005, group effect *p* = 0.530
	7 (ACI)	Mean MOCART border zone integration	3 months: 2.14 ± 0.38, 12 months: 2.86 ± 0.38, 24 months: 3.14 ± 0.38	Time effect *p* = 0.006, group effect *p* = 0.530
Bhattacharjee et al., 2016 [44]	11	Overall mean MOCART	61.0 ± 22.0	Not assessed
		MOCART border zone integration	Complete integration: 7/11 (63.6%), incomplete integration: 1/11 (9.10%), defect visible > 50% of length: 3/11 (27.3%)	Not assessed
	5	Integration parameter of ICRS II Histology Score (Scores 0–10, higher score means higher quality)	Patient 1: 11 months post-operation: 7.85, 24 months: 9.50 Patient 2: 24 months: 9.90 Patient 3: 13 months: 9.80, 37 months: 10.0 Patient 4: 12 months: 9.95 Patient 5: 18 months: 9.20	Not assessed
Ebert et al., 2017 [49]	31	Overall mean MOCART	3 months: 2.74 ± 0.10, 1 year: 3.11 ± 0.12, 2 years 3.22 ± 0.13, 5 years (*n* = 30): 3.14 ± 0.14	Time effect *p* = 0.028
		Mean MOCART border zone integration	3 months: 2.71 ± 0.20, 1 year: 3.00 ± 0.20, 2 years 3.16 ± 0.20, 5 years (*n* = 30): 3.10 ± 0.19	Time effect *p* = 0.38
Ebert et al., 2017 [54]	17 (ACI and 6-week return to full weight-bearing)	Overall mean MOCART	3 months: 2.97 ± 0.11, 12 months: 3.32 ± 0.16, 24 months: 3.46 ± 0.15	Time effect *p* < 0.0001, group effect *p* = 0.052, interaction effect *p* = 0.376
		Mean MOCART border zone integration	3 months: 2.88 ± 0.21, 12 months: 3.24 ± 0.23, 24 months: 3.29 ± 0.25	Time effect *p* = 0.062, group effect *p* = 0.041, interaction effect *p* =0.983
	14 (ACI and 8-week return to full weight-bearing)	Overall mean MOCART	3 months: 2.68 ± 0.11, 12 months: 3.01 ± 0.15, 24 months: 3.00 ± 0.15	Time effect *p* < 0.0001, group effect *p* = 0.052, interaction effect *p* = 0.376
		Mean MOCART border zone integration	3 months: 2.37 ± 0.19, 12 months: 2.79 ± 0.22, 24 months: 2.79 ± 0.23	Time effect *p* = 0.062, group effect *p* = 0.041, interaction effect *p* =0.983
Ogura et al., 2019 [45]	10	MOCART border zone integration	Complete integration: 8/10 (80.0%), incomplete integration: 1/10 (10.0%), defect visible < 50% of length: 1/10 (10.0%), defect visible > 50% of length: 0/10 (0.0%)	Not assessed
Yoon et al., 2020 [46]	10	Overall mean MOCART	3 months: 59.5 ± 29.4, 6 months: 65.5 ± 24.3, 12 months: 83 ± 11.1, 24 months: 85.0 ± 8.0	Not assessed
		MOCART border zone integration	3 months: Complete integration: 7/10 (70.0%), demarcating border: 1/10 (10.0%), defect visible < 50% of length: 1/10 (10.0%), defect visible > 50% of length: 1/10 (10.0%)	Not assessed
			6 months (*n* = 11 reported): Complete integration: 9/10 (90.0%), demarcating border: 0/10 (0.0%), defect visible < 50% of length: 1/10 (10.0%), defect visible > 50% of length: 1/10 (10.0%)	
			12 months: Complete integration: 10/10 (100.0%), demarcating border: 0/10 (0.0%), defect visible < 50% of length: 0/10 (0.0%), defect visible > 50% of length: 0/10 (0.0%)	
			24 months: 12 months: Complete integration: 10/10 (100.0%), demarcating border: 0/10 (0.0%), defect visible < 50% of length: 0/10 (0.0%), defect visible > 50% of length: 0/10 (0.0%)	
Słynarski et al., 2020 [48]	40	MOCART border zone integration	3 months: Observer 1: Complete integration: 29/38 (76.3%), demarcating border: 7/38 (18.4%), defect visible < 50% of length: 1/38 (2.6%), defect visible > 50% of length: 1/38 (2.6%) Observer 2: Complete integration: 38/40 (95.0%), demarcating border: 7/38 (18.4%), incomplete integration: 2/40 (5.0%)	Not assessed
			6 months: Observer 1: Complete integration: 22/38 (57.9%), demarcating border: 12/38 (31.6%), defect visible < 50% of length: 3/38 (7.9%), defect visible > 50% of length: 1/38 (2.6%) Observer 2: Complete integration: 35/38 (92.1%), demarcating border: 3/38 (7.9%), defect visible < 50% of length: 0/38 (0.0%), defect visible > 50% of length: 0/38 (0.0%)	
			12 months: Observer 1: Complete integration: 20/37 (54.1%), demarcating border: 17/38 (45.9%), defect visible < 50% of length: 0/38 (0.0%), defect visible > 50% of length: 0/38 (0.0%) Observer 2: Complete integration: 34/37 (91.9%), demarcating border: 2/37 (5.4%), defect visible < 50% of length: 1/37 (2.7%), defect visible > 50% of length: 0/38 (0.0%)	Not assessed
			24 months: Observer 1: Complete integration: 18/32 (56.3%), demarcating border: 13/32 (40.6%), defect visible < 50% of length: 1/32 (3.1%), defect visible > 50% of length: 0/38 (0.0%) Observer 2: Complete integration: 31/31 (100.0%), demarcating border: 0/31 (0.0%), defect visible < 50% of length: 0/31 (0.0%), defect visible > 50% of length: 0/31 (0.0%)	
	31 specimens (2 at 6 months, 27 at 12 months, 2 at 24 months)	Based on Histological ICRS II and O’Driscoll grading scales: Tidemark formation between implant and native cartilage	Interface absent: 17/29 (58.6%), Calcification front: 12/29 (41.4%), Single tidemark: 0/29 (0.0%)	Not assessed
Barié et al., 2020 [41]	9 (MACI)	Overall mean MOCART	58.9 ± 18.3	Overall mean MOCART: MACI vs ACI-P *p* = 0.206, Border integration: MACI vs ACI-P *p* = 0.206
		MOCART border zone integration	Complete integration: 2/9 (22.2%), incomplete integration: 2/9 (22.2%), <50% of length: 3/9 (33.4%), >50% of length: 2/9 (22.2%)	
	7 (ACI-P)	Overall mean MOCART	71.4 ± 19.3	
		MOCART border zone integration	Complete integration: 3/7 (42.8%), incomplete integration: 2/7 (28.6%), <50% of length: 2/7 (28.6%), >50 of length: 0/7 (0.0%).	

Abbreviations: SE, standard error; *n*, number of participants; MOCART, Magnetic Resonance Observation of Cartilage Repair Tissue; ICRS, International Cartilage Repair Society. Results are from the latest follow-up unless specified. MOCART border zone integration scoring system: complete, incomplete (split like border visible), defect visible <50% of length, defect visible >50% of length; Mean MOCART border zone integration scoring system: 1 = poor, 2 = fair, 3 = good, 4 = excellent.

**Table 4 ijms-23-04065-t004:** Subgroup meta-analyses on the use of collagen as a scaffold.

	Number of Cohorts	Number of Patients	Proportion or SMD	95% Confidence Interval	95% Prediction Interval	p_subgroup_
**Integration**						
Complete Integration						
Collagen Scaffold	4	69	0.7263	0.5361, 0.8591	0.4630, 0.8909	0.0616
Other techniques	8	154	0.5590	0.3859, 0.7189	0.1801, 0.8797	
**Clinical Scores**						
IKDC						
Collagen Scaffold	2	100	1.9780	1.0121, 2.9439	N/A	**0.0047**
Other techniques	8	169	1.4283	1.0051, 1.8516	0.3801, 2.4766	
KOOS Pain						
Collagen Scaffold	7	174	6.3287	4.1189, 8.5385	0.2874, 12.3700	**0.0064**
Other techniques	2	50	2.4036	−11.8279, 16.6351	N/A	
KOOS Symptoms						
Collagen Scaffold	7	174	5.5559	3.3265, 7.7854	−0.6257, 11.7376	0.0579
Other techniques	2	50	2.1616	−17.4148, 21.7380	N/A	
Lysholm						
Collagen Scaffold	2	57	3.6742	−12.6766, 20.0249	N/A	0.2645
Other techniques	5	54	1.8202	−1.0972, 4.7377	−6.1101 9.7506	
TAS						
Collagen Scaffold	4	71	3.2418	0.5963, 5.8872	−4.0359, 10.5194	**0.0090**
Other techniques	5	99	0.8512	−0.2112, 1.9135	−1.7837, 3.4860	
SF-36 Physical						
Collagen Scaffold	5	101	4.7544	1.9547, 7.5541	−2.7818, 12.2906	**0.0004**
Other techniques	2	16	1.1053	−1.7924, 4.0030	N/A	
SF-36 Mental						
Collagen Scaffold	5	101	2.2829	1.0191, 3.5467	−0.9880, 5.5538	**<0.0001**
Other techniques	2	16	−0.5795	−3.3890, 2.2300	N/A	
**Graft Failure**						
Collagen Scaffold	8	282	0.0657	0.0442, 0.0965	0.0436, 0.0977	0.3449
Other techniques	4	61	0.0904	0.0345, 0.2162	0.0179, 0.3513	

## Data Availability

The data are contained within the article and Appendix A.

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
