# Peer review of "The Use of Autologous Chondrocyte and Mesenchymal Stem Cell Implants for the Treatment of Focal Chondral Defects in Human Knee Joints—A Systematic Review and Meta-Analysis"

_ijms, 2022, doi:10.3390/ijms23074065_

Round 1
Reviewer 1 Report
Dear Authors,
from your manuscript as well as from Migliorini's recent review (Migliorini et al. Journal of Orthopaedic Surgery and Research. 2021.16:543) it is clear that most papers, investigating chondral procedures augmented with MSCs, referred often to a limited length of the follow-up, and the cell delivery protocol are heterogeneous.
I suggest implementing the discussion (lines 458-461 and lines 568-575) with the following bibliographical references:
- Cavinatto L, Hinckel BB, Tomlinson RE, Gupta S, Farr J, Bartolozzi AR. The role of bone marrow aspirate concentrate for the treatment of focal chondral lesions of the knee: a systematic review and critical analysis of animal and clinical studies. Arthroscopy. 2019;35(6):1860–77.
- Filardo G, Madry H, Jelic M, Roffi A, Cucchiarini M, Kon E. Mesenchymal
stem cells for the treatment of cartilage lesions: from preclinical findings to clinical application in orthopaedics. Knee Surg Sports Traumatol Arthrosc. 2013;21(8):1717–29. - Brittberg M. Cell carriers as the next generation of cell therapy for cartilage repair: a review of the matrix-induced autologous chondrocyte implantation procedure. Am J Sports Med. 2010;38(6):1259–71.
- Makris EA, Gomoll AH, Malizos KN, Hu JC, Athanasiou KA. Repair and tissue engineering techniques for articular cartilage. Nat Rev Rheumatol. 2015; 11(1):21–34.
Author Response
Dear Reviewer,
Thank you very much for taking the time to review our work. We are grateful for your recommendations and believe this has greatly improved the discussion section of our review.
In accordance with your suggestion for revisions, we have included the citations you have recommended. These are referenced in the areas you had brought to our attention, which now correspond to lines 470-476 and lines 601-637. We have briefly mentioned the issues you highlighted in the first area, with the aim to explore these further in the discussion which follows. In the second instance, more in-depth evaluations of the implications of study heterogeneity and the lack of long-term follow-up data are discussed, with reference to the articles you recommended. These articles now correspond to reference numbers 56, 57, 58 and 65.
Thank you, once again, for your time and valuable recommendations.
Reviewer 2 Report
This review reports the quality of integration of the autologous and mesenchymal stem cell implants to treat focal defects of cartilage. This is a serious and dense work from the authors. Clinical studies only were selected, which is a strong point. The limitations are carefully reported, in particular the absence of the quality control of the phenotype of the cells that are used to make the implants. This review is original and brings important and new information to the field of tissue engineering of cartilage. This is a serious and dense work from the authors.
Comments: please correct numerous typographical errors ( eg in the discussion : lines 539, 541 546 548...).
Figure 7: the text is not easily readable.
Author Response
Dear Reviewer,
Thank you very much for your encouraging comments. We are pleased that you found the piece valuable and are very grateful that you took the time to review our work.
Our responses to your recommendations for revisions are as follows:
- Typographical errors in the lines mentioned and other areas have now been corrected.
- Figure 7 has been reproduced to have larger and clearer text.
Thank you, once again, for your time and insight.